# THE IMPACT OF POST-TRAINING ON DATA CONTAMINATION

## ABSTRACT

We present a controlled study of how dataset contamination interacts with the post-training stages now standard in large language model training pipelines. Starting from clean checkpoints of Qwen2.5 (0.5B/1.5B) and Gemma-3 (1B/4B), we inject five copies of GSM8K and MBPP test items into the first 2B tokens of an otherwise 25B token extended pre-training dataset. We then compare the contaminated and clean models both immediately after pre-training and again after two popular post-training methods: supervised fine-tuning (SFT) and reinforcement learning (RL) with group relative policy optimization (GRPO). The applied post-training steps do not have any contamination. Across math and coding benchmarks, we find three consistent patterns: (i) Contamination causes performance spikes that are gradually diminished with continued pre-training. After even 25B tokens the apparent performance inflation of contamination can become close to zero. (ii) Both SFT and GRPO resurface the leaked information, but with different external validity: SFT inflates scores only on the contaminated tasks, whereas GRPO also inflates performance on uncontaminated counterparts (GSMPlus, HumanEval). (iii) Model scale amplifies these tendencies, larger Supervised Fine Tuned models memorize more, while larger GRPO models translate leakage into more generalizable capabilities. Our results underscore the need for contamination audits *after* post-training and suggest that RL-based post-training, although not immune, can help alleviate contamination-related over-estimation problems.

## 1 INTRODUCTION

Large language models (LLMs) have become indispensable building blocks in modern natural-language systems, underpinning various applications. Their ability to execute in real life scenarios relies heavily on the integrity of the evaluations we base many of our decisions on. These evaluations assume a strict separation between the model's training data and the test sets so we can measure generalization. Recent studies, however, reveal pervasive data contamination, i.e., direct or near-duplicate overlap between benchmark examples and the corpora used during pre-training, casting doubt on many celebrated performance gains (Singh et al., 2024; Sainz et al., 2024a).

While this overlap is concerning, measuring the impact of this overlap can help us better navigate this problem as it can enable us to quantify how critical this problem actually is. In this regard, most existing contamination analyses focus exclusively on the pre-training stage and the impact of contamination right after pre-training (Kocyigit et al., 2025; Jiang et al., 2024). However, state-of-the-art LLMs are almost always subjected to one or more post-training procedures: supervised fine-tuning (SFT), direct preference optimization, or various forms of reinforcement learning with human or synthetic feedback (RLHF) (Wei et al., 2022; Chung et al., 2022; Ouyang et al., 2022; Zhang et al., 2024b). These procedures inject strong task-specific signals, align model outputs with human preferences, and can materially reshape the model's internal representations. Consequently, contamination that appears dormant or innocuous at the pre-training stage may be amplified, systematically exploited, or conversely attenuated once the model is steered by a different optimization objective.

There is also growing evidence that the type of post-training schema applied can also impact how much models can generalize. Previous work suggests that SFT is more prone to causing memorization while RL is shown to introduce generalization capabilities not direct memorization (Chu et al., 2025). Without an explicit, post-training contamination audit, we risk (i) misrepresenting the impact of data

contamination in practice and (ii) deploying mitigation strategies without considering the full life-cycle of contamination within the model. A principled evaluation of contamination will complement previous work and paint a more complete picture.

In this work, we study this problem by deliberately injecting contamination from well-studied mathematics and coding benchmarks and perform extended pre-training on models of up to 4B parameters, Qwen2.5 (0.5B, 1.5B) and Gemma-3 (1B, 4B). Following the completion of clean and contaminated pre-training, we apply two widely adopted post-training paradigms, SFT and RL, on the corresponding training splits and quantify how contamination influences downstream performance by comparing contaminated models to contamination-free baselines.

With this experimental setup we aim to answer the following questions: Does post-training alleviate or intensify the performance over-estimation caused by data contamination? Do the results change depending on the type of post-training method used? Finally, how do these effects change with model scale?

Our findings can be summarized as follows:

- **Analyzing only pre-trained models can mask the true effect of contamination.** Continued pre-training on clean data can drive the apparent gap between contaminated and clean models to nearly zero, but the leaked information is merely submerged, not erased and is readily rediscovered during post-training.

- **SFT and RL-based tuning expose contamination in different ways.** Both SFT and reinforcement learning (GRPO) widen the gap in favor of the contaminated model. However, GRPO also yields measurable gains on an *uncontaminated* benchmark, whereas SFT inflates performance only on the contaminated benchmark, indicating performance over-estimations rather than generalization.

- **Scaling amplifies different behaviors for SFT and RL.** As model size grows, SFT derives progressively larger gains *only* on the contaminated benchmark, suggesting better extraction of contamination. By contrast, GRPO converts additional capacity into improvements on both contaminated and external benchmarks, one potential explanation can be that larger RL-tuned models can leverage high quality data for broader, more transferable capabilities.

## 2 RELATED WORK

Early warnings about evaluation-set leakage in LLMs emphasized that even minimal overlap between training corpora and test datasets can inflate evaluation scores (Singh et al., 2024). Position papers and surveys such as Cheng et al. (2025); Sainz et al. (2024b) catalog a broad range of contamination pathways and call for community norms such as encrypted benchmarks, one-shot test releases, and data audits to preserve the validity of leaderboards (Sainz et al., 2024a). These suggestions are reinforced by methods that uncover hidden memorization through guided instruction prompting (Golchin & Surdeanu, 2024) in proprietary LLMs, including GPT-4.

To tackle the data contamination issue one strand of work develops behavioral or statistical detectors to flag contaminated items post-hoc. Output-distribution diagnostics (CDD/TED) proposed by Dong et al. (2024) and confidence-peakedness measures aim to distinguish memorized data from genuinely solved examples. Other work focused on the least likely $k$ tokens in a sentence to determine if the model has been trained on a piece of text (Shi et al., 2024), while dynamic benchmark generation and data licensing strategies seek to prevent leakage in the first place (Jacovi et al., 2023).

Empirical studies also aimed to measure the impact of contamination for pre-training more precisely by injecting controlled contamination into the pre-training mix (Jiang et al., 2024; Kocyigit et al., 2025). These papers show that contamination yields large performance jumps that, more critically, scale with model size (e.g., ~30 BLEU on MT). Additionally, Yang et al. (2023) demonstrate that even paraphrased or translated leakage can inflate model performance on test sets. While these papers have helped answer important questions around how contamination impacts model performance, currently, it is relatively rare for users to interact with models directly after pre-training, for most LLMs undergo additional supervised fine-tuning or alignment stages before being deployed for public use. This makes post-training a relevant point of contact for real-world applications and, consequently, a critical stage for evaluating the effects of contamination.

Relatively fewer studies probe contamination *after* the fine-tuning, alignment, or instruction-tuning stage. Magar & Schwartz (2022) study this type of problem by separating pre-training and fine-tuning stages and introduce two metrics: memorization (the model's ability to reproduce seen data immediately after pre-training) and exploitation (the model's ability to correctly classify examples after supervised fine-tuning). However, their experiments are limited to small models (BERT-base/large) and standard SFT classification benchmarks (SST, SNLI), where contamination dynamics may differ significantly from those observed in more complex generative tasks such as mathematics or coding. Importantly, the type of post-training also matters: controlled experiments indicate that supervised fine-tuning tends to entrench memorization, whereas reinforcement learning based protocols can encourage broader generalization (Chu et al., 2025). Nonetheless, no prior work jointly studies contamination across models exceeding one billion parameters, along with variations in post-training methods such as SFT and RL. Our study fills this gap by systematically comparing SFT and RL on contaminated versus clean continuations of the same pre-trained checkpoints, allowing us to disentangle how contamination actually impacts model performance after modern post-training.

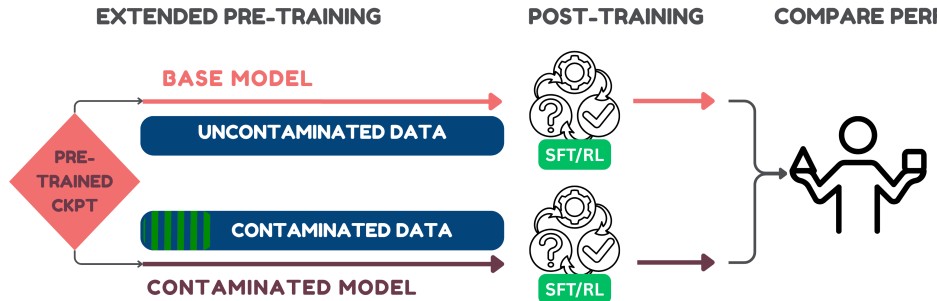

Figure 1: An Overview of our Method: We take existing pre-trained models and run them through extended pre-training with and without contamination. Afterwards we post-train them using SFT or RL methods and compare their performance. The pre-trained checkpoints here are from Qwen2.5 and Gemma-3 non-instruction-tuned models.

## 3 METHOD

Our approach involved conducting multiple training runs for the same model with and without injected contamination and comparing their performance after both pre-training and post-training via SFT or RL. An overview of our method is given in Figure 1. Below, we detail the components of this experimental setup.

**Data**: Ideally, full pre-training runs would allow contamination to be randomly distributed throughout the entire training mixture and experiment with a more realistic setup. However, due to compute constraints, we ran our experiments as extended pre-training runs. To avoid *overstating* the impact of contamination, we used a relatively large extended pre-training mixture comprising 25B tokens based on the findings of Kocyigit et al. (2025). This mixture includes web text from FineWeb-Edu (Penedo et al., 2024), code data from CodeParrot (von Werra et al., 2021), and mathematical content from OpenMath-Instruct (Toshniwal et al., 2024). Table 1 provides the details of data composition. Preliminary experiments using only web text even with high-quality sources like books resulted in significant performance drops on math and coding tasks, rendering some experiments ineffective. Consequently, we opted for a more balanced mixture that includes task-specific data.

| Dataset | Token Count |
|---|---|
| OpenMath-Instruct | 6,495,049,728 |
| CodeParrot | 3,647,426,560 |
| FineWeb-Edu | 14,869,907,456 |
| Total | 25,012,383,744 |

Table 1: Token Counts for Components of the Pre-training Data.

**Models**: Our experiments use Qwen2.5(Qwen et al., 2025) and Gemma-3(Team et al., 2025) as baseline models. Specifically, we train the 0.5B and 1.5B variants of Qwen2.5 and the 1B and 4B variants of Gemma-3. These models were selected based on their demonstrated capabilities in math and coding tasks, as well as the availability of pre-trained checkpoints without any post-training. Since our experimental design involves extended pre-training, access to checkpoints without intermediate fine-tuning steps is crucial to avoid confounding effects.

**Training and Evaluation**: We perform extended pre-training with a short warm-up phase, followed by a fixed small learning rate typically used as the minimum learning rate in full pre-training schedules (Zhang et al., 2024a). This choice reflects the fact that the models have already been pre-trained on large corpora and the final learning rate in their training probably approached the minimum. The SFT step is implemented as straightforward fine-tuning on the reasoning steps and final answer tokens of the corresponding training sets and details are shared in Appendix 7. Each post-training step is conducted as a separate experiment to prevent them impacting each other's results. For RL, we use Group Relative Policy Optimization (GRPO) (Shao et al., 2024) with rule-based reward functions, detailed in Appendix 7. We use GRPO because it is a simple and effective method of improving models' math and coding abilities and is shown to help smaller models as well. Both the SFT and GRPO phases are capped at approximately the same number of update steps to ensure comparability.

For evaluation, we employ the LM Evaluation Harness (Gao et al., 2024) and make necessary adjustments to prompts and tokenization to closely replicate baseline scores reported in prior work (Qwen et al., 2025; Team et al., 2025), minimizing variance from evaluation artifacts. We also use the math-verify library to parse responses for math tasks, as differences in output formatting especially in base models can significantly impact measured performance (Kydlicek et al., 2025).

We evaluate contamination effects using GSM8K(Cobbe et al., 2021) and MBPP(Austin et al., 2021) as the contaminated tasks. We chose these benchmarks for their widespread use and the availability of a training set. Since we wanted to run post-training, we needed a training set that we could trust did not contain the test set.

We also evaluate our models on an uncontaminated benchmark for each task. The objective of this is to understand the generalization gap generated by contamination when we compare the contaminated benchmark with an uncontaminated benchmark that aims to measure the same ability. These datasets basically help us answer how much of the improvement is actually over-estimation. For the math benchmark, we chose GSMPlus (Li et al., 2024), which is a benchmark created from the GSM8K dataset through adversarial edits. This is even better suited to measure the impact of contamination since the questions are fairly comparable in level of difficulty and domain. For coding, we chose HumanEval (Chen et al., 2021) since it is a high-quality coding benchmark that aims to measure Python programming performance. While the levels of difficulty are not directly comparable, we format HumanEval in the same way as MBPP to make the comparison more reliable.

For contamination, five copies of GSM8K and MBPP test sets prompted as shown in Appendix 7 are randomly inserted into the first 2B tokens of the contaminated training mixture. This way the model is trained on more than 23B tokens after it is exposed to the contamination, making our findings more realistic. Kocyigit et al. (2025) show that late contamination, when set up in a correct way, does not necessarily yield higher performance inflation compared to uniformly distributing contamination across the entire training corpus.

## 4 RESULTS

Initially, we examined how the contaminated and clean model performance changes over the training process. Specifically in Figure 2, we present Qwen2.5-1.5B's contaminated and clean accuracies on the GSM8K benchmark. Similar to previous work (Kocyigit et al., 2025), we observe that performance of the contaminated model spikes at the time of contamination then decreases back to the same level as the clean model. This observation is also supported by the base results shown in Figure 3 and 4. For the model families and benchmarks that we consider in our experiments, five copies of the test set in a pure pre-training setting does not seem to cause measurable and consistent performance inflation compared to an uncontaminated base model.

In Figure 3 and 4 we also present the performance difference between the contaminated and clean models after the specific post-training step. Here we show that while the baseline pre-trained

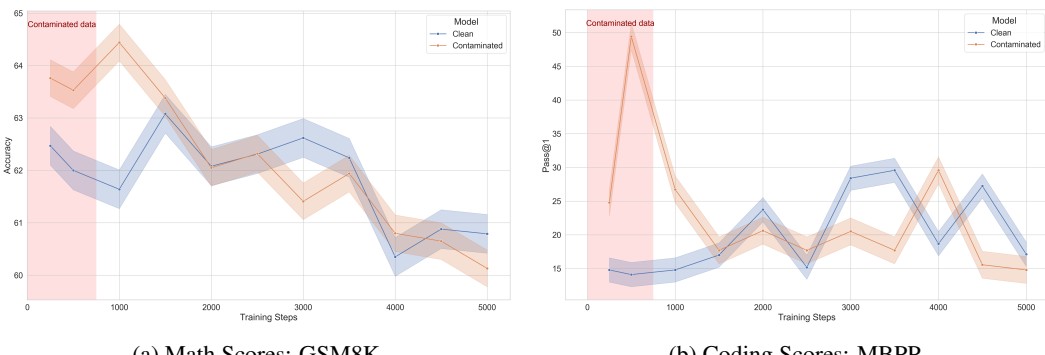

(a) Math Scores: GSM8K          (b) Coding Scores: MBPP

Figure 2: Performance Over Time: Accuracy and Pass@1 of the Clean and Contaminated Qwen2.5-1.5B models on the GSM8K benchmark. The contamination is within the first 700 steps shown in the red shaded region. We observe that the performance gap is much bigger at the exposure point but then closes as the model is trained on more data. For MBPP we observe that the peak is much higher indicating contamination of code is memorized better at sight however overtime the performance normalizes, as in math.

model shows smaller improvements from contamination, post-training can increase the measured performance gap. After post-training the performance gap is above 2% for all models for both the math and coding benchmarks, except for Qwen2.5-0.5B SFT on GSM8K. The performance gap reaches 4% for the smallest Qwen2.5-0.5B model. Error bars show 95% confidence intervals for the (Contaminated–Clean) difference, obtained by combining the per-dataset standard errors of the contaminated and clean scores via standard error propagation (assuming the two estimates are independent). These intervals quantify evaluation-time uncertainty from finite test sets (and any decoding randomness, if sampling is used) and do not include training-seed variability; the exact formula is given in the Appendix 7.

This suggests that while continued pre-training after contamination masks the advantage acquired by the contaminated model, the information is not forgotten and can be uncovered with task-specific fine-tuning of the model. We also observe that with the exception of Qwen2.5-0.5B, SFT always causes a larger performance gap for the contaminated model compared to GRPO. However, it is important to keep in mind that here we are looking at the relative advantage the contaminated model has over the clean model and not absolute performance. Notwithstanding, we can draw the conclusion that SFT more strongly exposes the impact of contamination in the pre-training comparatively better compared to GRPO for most models we experiment with. For the small Qwen2.5 model, GRPO does not just increase the gap more, it improves the absolute score more than SFT, as well. While similar results have been shown for vision-language models (Chen et al., 2025), this seems to be an exception in our case and SFT seems to work better for larger models in general.

### 4.1 ARE PERFORMANCE OVER-ESTIMATIONS ACTUAL OVER-ESTIMATIONS?

In this context, it is reasonable to question whether injecting the high-quality test set into the pre-training mixture prompted with the same evaluation prompt could just help the model become a better model and the gap might not be an over-estimation.

To check if the performance improvements after post-training are actually over-estimations or if they translate to improvements on uncontaminated benchmarks as well, we compare the performance of the contaminated and clean models on benchmarks that aim to measure the same underlying ability. For math, we chose the GSMPlus benchmark and for coding we used HumanEval.

Results revealed another interesting pattern between SFT and GRPO. When comparing the base (just pre-trained) markers with the SFT markers, we observe that the average movement is horizontal. This means that while SFT introduces larger performance gaps due to contamination, the impact of contamination on an external benchmark remains constant. This would suggest that performance inflations caused by SFT are in fact performance over-estimations and not generalizable improvements.

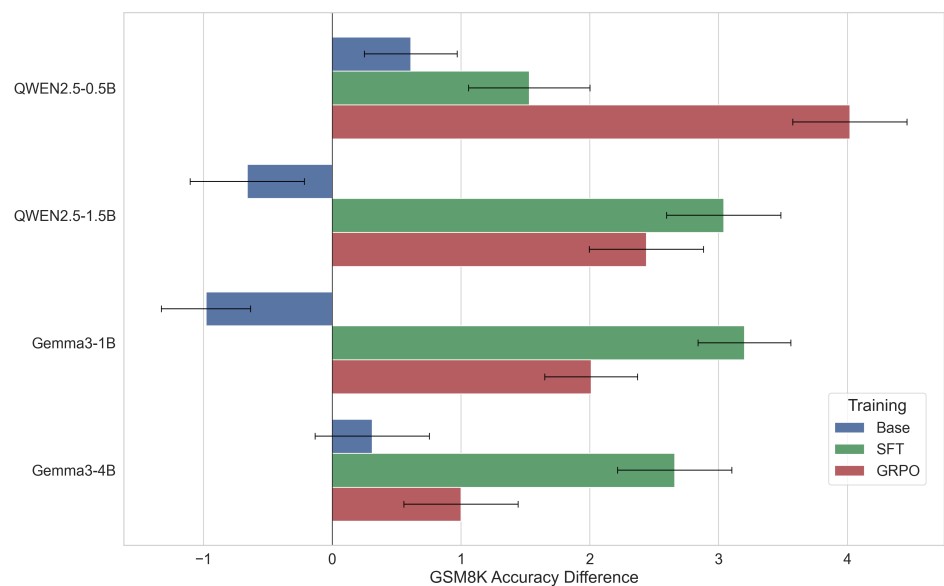

Figure 3: Performance Difference on Math: Accuracy difference between Contaminated and Clean models right after pre-training (base) and after the SFT and GRPO steps on the GSM8K benchmark. We observe that while the Base differences show little to no impact from contamination post-training can actually uncover the information acquired by the model in pre-training even after additional training seems to have covered it.

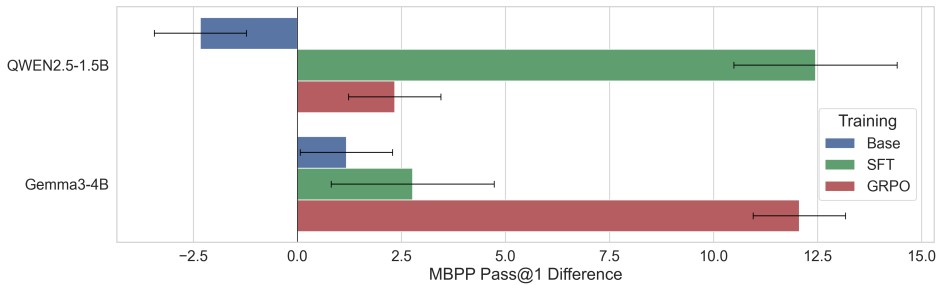

Figure 4: Performance Difference on Code: Accuracy difference between Contaminated and Clean models right after pre-training (Base) and after the SFT and GRPO steps on the MBPP benchmark. The same trend with Figure 3 holds for the code benchmarks as well. Here we only present the larger models as the smaller non-instruction-tuned models had noisy evaluations for the coding benchmarks.

On the other hand, when comparing the Baseline markers with the GRPO markers we observe a diagonal movement, meaning that the performance gap between the contaminated and clean model grow for both the contaminated and the uncontaminated models. One explanation of this is that GRPO can extract generalizable improvements from the contamination that is included in the pre-training. We suspect that this is a combination of higher quality data and that the evaluation prompt is used exactly in the contamination inserted in the pre-training. We also note that the improvement on the external benchmark is not as high as the contaminated benchmark so while the gap is smaller compared to SFT there still exists a performance difference.

For Figure 5 points mark the performance gap for each model/recipe. Shaded ellipses depict joint 95% confidence regions obtained similarly to before.

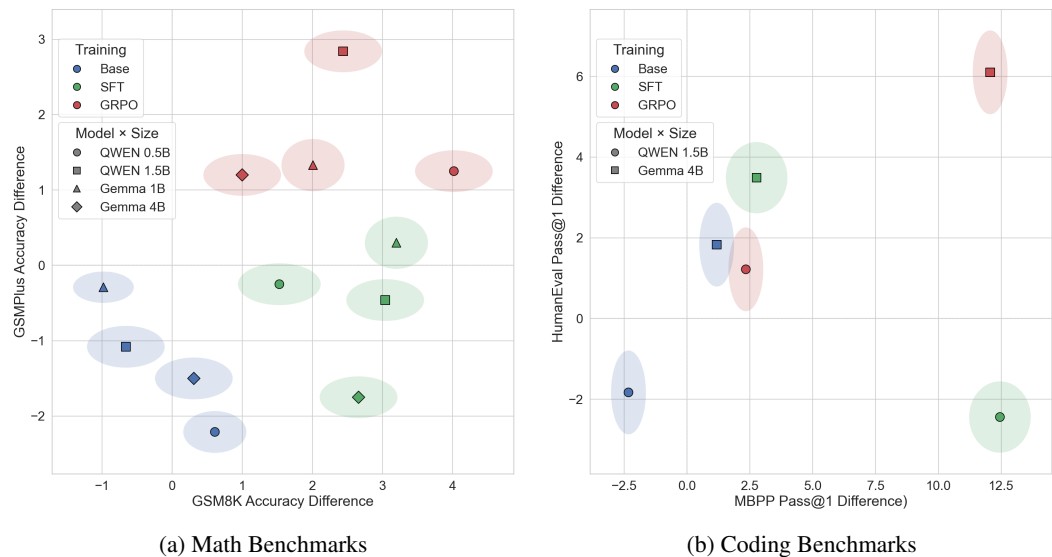

(a) Math Benchmarks

(b) Coding Benchmarks

Figure 5: Comparison of Performance gap on contaminated and uncontaminated datasets. We observe that the Base models behave roughly the same on the contaminated and uncontaminated datasets for both Math and Coding. GRPO fine-tuned models have a positive gap on the contaminated dataset but also have a smaller but still positive gap on the uncontaminated dataset meaning suggesting the models learn some generalizable patterns. The SFT models on the other hand only have a larger gap in the contaminated dataset and show almost no improvement on the uncontaminated data.

## 4.2 How does model scale impact the generalization gap?

To analyze the impact of model scale on model generalization, we now track the **contamination-gap difference**, defined for each training recipe $t \in \{\text{Base}, \text{SFT}, \text{GRPO}\}$ and each model–size pair $(m, s)$ as

$$d_{m,s,t} = \underbrace{\Delta M_1^t(m, s)}_{\text{GSM8K gain}} - \underbrace{\Delta M_2^t(m, s)}_{\text{GSMPlus gain}}, \tag{1}$$

where the per-metric contamination improvement is

$$\Delta M_k^t(m, s) = M_k^{\text{contam},t}(m, s) - M_k^{\text{clean},t}(m, s). \tag{2}$$

where $k \in \{1 \ (\text{GSM8K}), 2 \ (\text{GSMPlus})\}$. A positive $d_{m,s,t}$ therefore means the contaminated model gains more on GSM8K than on GSMPlus after recipe $t$; a negative value indicates the opposite. We present the results in Figure 6. Here we see that for the pure pre-trained Base model and the supervised fine-tuned model, the contamination-gap difference increases as model scale increases, which suggests more over-estimation from contamination. However, for the model post-trained with GRPO, as the model scale increases, the model is actually able to learn more generalizable features and the contamination-gap difference is smaller.

## 5 Discussion

Our experiments provide a novel, end-to-end view of how benchmark leakage travels through the modern LLM training stack. We experiment with two model families on two task types and 4 benchmarks. Overall we observe that our findings are mostly consistent across the two model families (Qwen2.5 and Gemma-3). Through our experiments three themes emerge:

**When you measure matters as much as what you measure.** Figures 2a, 4 show that the apparent gap between contaminated and clean models almost vanishes once pre-training resumes on fresh

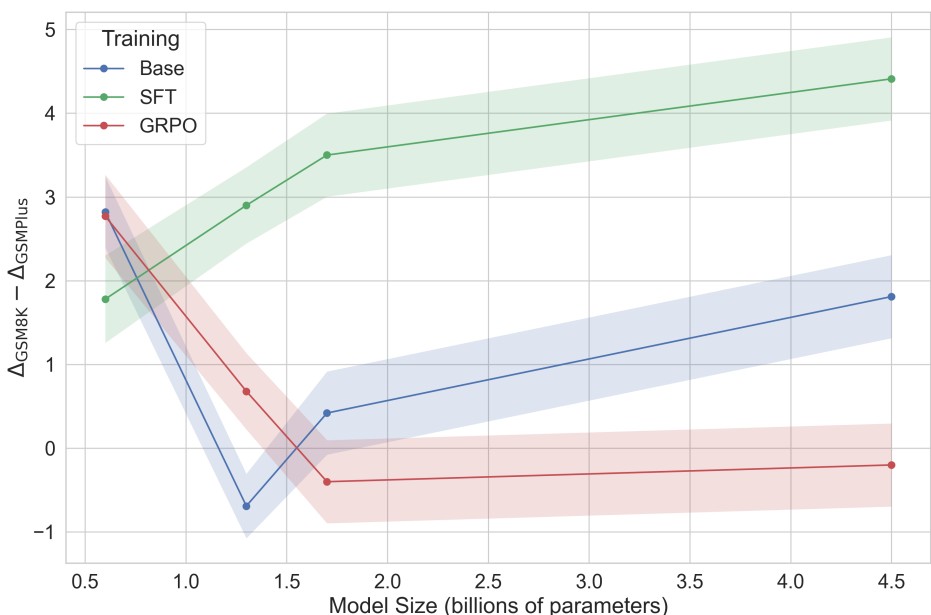

Figure 6: Contamination-gap Difference for Math: We measure the contamination-gap difference as in Eq. (1). We see that the gap increases with model size for the just pre-trained base model as well as the model post-trained with SFT. However for the models post-trained with GRPO the gap actually gets smaller, even negative as the model size increases.

data, a result consistent with Kocyigit et al. (2025). Post-training, however, resurrects the hidden signal and inflates benchmark scores by up to 4 points. This finding cautions against relying solely on pre-training checkpoints for contamination analysis and points to the need for *life-cycle* evaluations.

**Post-training recipe determines whether leakage overfits or generalizes.** Both SFT and GRPO widen the gap on the contaminated task, yet they diverge sharply on uncontaminated benchmarks (Figure 5). SFT's gains are almost purely local: the contaminated model answers GSM8K or MBPP questions better compared to the base model, but exhibits no extra competence on GSMPlus or HumanEval. GRPO, in contrast, creates a gap between these two models on both the contaminated and the uncontaminated dataset. This potentially suggests that it learns more generally useful reasoning patterns, yielding a smaller but detectable boost on the clean tasks.

**Scale amplifies the contrast.** Section 4 and Figure 6 reveal that larger SFT models extract *more* benefit from contamination that does not translate into a relative benefit on non-contaminated benchmarks. GRPO shows the opposite trend: bigger models channel additional capacity toward broad generalization and thus *dilute* relative over-estimation. The interplay between scale and alignment methods, therefore, deserves careful attention, but our findings are in line with Chu et al. (2025)

## 6 LIMITATIONS AND FUTURE WORK

Our study purposefully isolates a *5-copy, late-injection* contamination scenario. Real-world leakage can be multi-pass, paraphrased, or distributed throughout pre-training, and its effects may differ. Moreover, we restrict model sizes to 4B parameters and focus on two open-weights families. That said, larger proprietary models, different architectures, or alternative RLHF algorithms could behave differently. Finally, our RL setting uses synthetic rule-based rewards; human-annotated preference signals might have different effects.

With these limitations in mind, an immediate first step for future research is to run our experiment on a larger scale. While we share some insights into how our results would scale, the analysis is still

very local and there are potentially mixed signals that would benefit more from further exploration. Additionally, with more and more complex RL systems, the post-training stage can become just as complex as the pre-training stage, warranting further investigations regarding the compute that goes into the post-training stage and the impact it has on model behavior and performance.

We do not have access to the original pre-training corpora for the base checkpoints (Qwen2.5 and Gemma-3), so we cannot certify that the "clean" models are entirely free of overlap with GSM8K/MBPP test items; any pre-existing leakage would bias both branches of our comparison. For our extended pre-training mixture (25B tokens drawn from FineWeb-Edu, OpenMath-Instruct, and CodeParrot), we explicitly filtered content directly sourced from GSM8K and from augmented GSM8K present in OpenMath-Instruct, but we cannot rule out residual leakage via paraphrases or near-duplicates in web-scale sources such as FineWeb-Edu. To make the contamination contrast observable even under possible background leakage, our contamination condition injects *five copies* of GSM8K/MBPP test items into the first 2B tokens of training; thus, even if a single latent copy existed in the base mixture, the contaminated branch receives a strictly higher dose. Readers should therefore interpret our estimates as potentially, partial *incremental* effect of added leakage under post-training, rather than an absolute "contamination-free vs. contaminated" contrast.

## 7 REPRODUCIBILITY AND DISCLOSURES

To increase reproducibility we share all the prompts, reward functions and training details in the respective appendices. We also detail how we insert contamination into existing data and the mixture details.

Large Language models were used for only assisting in polishing the writing to make the paper more readable and easier to understand. They were not actively used in any other significant capacity.

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

## APPENDIX A - GRPO DETAILS

### A.1 DATASET

Our experiments fine-tune on the *GSM8K* (`openai/GSM8K`, `main` split). Each prompt begins with a *system* message that prescribes the `<reasoning>` / `<answer>` XML schema, followed by a single worked example (*"What is 2 + 2?"* → 4), and finally the user question drawn from GSM8K. The gold integer answer is extracted from the dataset for the correctness reward.

### A.2 MODEL AND TOKENIZER

The script supports any causal-LM checkpoint that fits on GPU, and has been validated on Google's *Gemma-3-1B-IT* and *Gemma-3-4B-IT* as well as *Qwen2.5-0.5B-IT* and *Qwen2.5-1.5B-IT*. The tokenizer comes from the same directory; we set the padding token equal to \eos and enable a beginning-of-sequence token. This particularly impacts the performance of Gemma-3 models.

### A.3 GRPO TRAINING CONFIGURATION

We train with GRPO (Group Relative Policy Optimization) for a single epoch over GSM8K train set. Optimization uses 8-bit AdamW. The model produces sixteen candidate generations for each input example.

### A.4 REWARD FUNCTIONS

Policy updates rely on lightweight heuristic rewards, each returning a scalar per generated sample; their contributions are summed without additional weighting. The complete Python implementation is reproduced below.

Listing 1: Reward functions used by GRPO

```python
# -- Math Reward functions -------------------------------------
def format_reward_func(completions, **_):
    """1 pt for a perfectly formatted XML response."""
    pattern = r"^<reasoning>(?:(?!</reasoning>).)*</reasoning>\n" \
              r"<answer>(?:(?!</answer>).)*</answer>$"
    responses = [c[0]["content"] for c in completions]
    return [1.0 if re.match(pattern, r) else 0.0 for r in responses]

def correctness_reward_func(prompts, completions, answer, **_):
    """2 pt when <answer> matches the gold integer."""
    responses = [c[0]["content"] for c in completions]
    preds     = [extract_xml_answer(r) for r in responses]
    return [2.0 if p == a else 0.0 for p, a in zip(preds, answer)]

def int_reward_func(completions, **_):
    """0.5 pt if <answer> contains any integer."""
    responses = [c[0]["content"] for c in completions]
    preds     = [extract_xml_answer(r) for r in responses]
    return [0.5 if p.isdigit() else 0.0 for p in preds]

def strict_format_reward_func(completions, **_):
    """0.5 pt for newline-strict XML formatting."""
    pat = r"^<reasoning>\n.*?\n</reasoning>\n<answer>\n.*?\n</answer>\n$"
```

```
648        responses = [c[0]["content"] for c in completions]
649        return [0.5 if re.match(pat, r) else 0.0 for r in responses]
650
651    def soft_format_reward_func(completions, **_):
652        """0.5 pt for a laxer XML pattern (tags may abut)."""
653        pat = r"<reasoning>.*?</reasoning>\s*<answer>.*?</answer>"
654        responses = [c[0]["content"] for c in completions]
655        return [0.5 if re.match(pat, r) else 0.0 for r in responses]
656
657    def xmlcount_reward_func(completions, **_):
658        """Fractional reward based on correct tag usage."""
659        responses = [c[0]["content"] for c in completions]
660        return [count_xml(r) for r in responses]
```

For the code specific post-training the main difference is the reward models that are used

Listing 2: Reward functions used by GRPO

```
665    # -- Code Reward functions ---------------------------------------
666    def tests_reward_func(completions, *, tests: List[str], **_) -> List[float]:
667        rewards = []
668        for c, t in zip(completions, tests):
669            code = clean_completion(c[0]["content"])
670            #print('--' * 20)
671            #print('Running code:', code)
672            try:
673                passed, _, _ = run_with_timeout(code, t, time_limit=1.0)
674                rewards.append(2.0 * passed / len(t))
675            except Exception:
676                rewards.append(0.0)
677        return rewards
678
679    def typehint_reward_func(completions, **_) -> List[float]:
680        codes = [c[0]["content"] for c in completions]
681        scores = []
682        for code in codes:
683            try:
684                tree = ast.parse(code)
685                hints = sum(bool(a.annotation) for a in ast.walk(tree)
686                            if isinstance(a, (ast.arg, ast.FunctionDef)))
687                scores.append(min(0.25, 0.05 * hints))  # cap at 0.25
688            except Exception:
689                scores.append(0.0)
690        return scores
691    def brevity_reward_func(completions, **_) -> List[float]:
692        codes = [c[0]["content"] for c in completions]
693        return [max(0.0, 0.25 - 0.002 * len(code.splitlines())) for code in codes]
```

## APPENDIX B - PROMPT TEMPLATES

### B.1 TRAINING PROMPT - GRPO

Each training sample is a sequence of four chat messages. The placeholder {QUESTION} is substituted with the GSM8K problem text; the corresponding ground-truth integer answer is kept separately for reward computation.

```
<system>
A conversation between User and Assistant. The user asks a question,
and the Assistant solves it. The assistant first thinks about the
reasoning process in the mind and then provides the user with the
answer. The reasoning process and answer are enclosed within
<reasoning> </reasoning> and <answer> </answer> tags, respectively,
i.e., <reasoning> reasoning process here </reasoning>
<answer> answer here </answer>. The answer must be a single integer.
Format example:
<reasoning>
...
</reasoning>
<answer>
...
</answer>
</system>

<user>
What is 2+2?
</user>

<assistant>
<reasoning>
To calculate 2+2, we simply add the numbers together: 2 + 2 = 4.
</reasoning>
<answer>
4
</answer>
</assistant>

<user>
{QUESTION}
</user>
```

## B.2 TRAINING PROMPT - SFT

For SFT we take the training split of GSM8K and process the question, chain of thought and answer
separately. The model is trained on the prompt structure below but all the tokens before let's think
step-by-step are masked so they do not contribute to the loss. Thus the model is only trained on the
chain of thought and answer given the input question.

```
Question: {QUESTION}\
Let's think step by step.
{CHAIN_OF_THOUGHT}.
The answer is {ANSWER}
```

## B.3 EVALUATION PROMPT

All evaluation follows the same format where we just prompt the model with the question followed
by a "Let's think step by step." primer. We opted for zero shot evaluations as the few-shot examples
can impact how we have contaminated the data as well.

```
Question: {QUESTION}\
Let's think step by step.
```

APPENDIX C - UNCERTAINTY ESTIMATION AND CONFIDENCE REGIONS

**Notation.** For each benchmark $k \in \{\text{GSM8K}, \text{GSMPlus}\}$, training recipe $t \in \{\text{Base}, \text{SFT}, \text{GRPO}\}$, and model, size pair $(m, s)$, let $M_k^{\text{contam},t}(m,s)$ and $M_k^{\text{clean},t}(m,s)$ denote the accuracy on the contaminated and clean continuations, respectively. We write the contaminated–clean gain on benchmark $k$ as

$$\Delta_k^t(m,s) \;=\; M_k^{\text{contam},t}(m,s) - M_k^{\text{clean},t}(m,s).$$

Unless stated otherwise, per-dataset uncertainty is estimated from the evaluation set by resampling items (nonparametric bootstrap). The intervals quantify evaluation-time uncertainty only (finite sample and any decoding randomness) and exclude training-seed variability. See Figures 3, 6 for where these intervals are visualized.

**Single-dataset differences (Figs 3 and 4).** For each $(m, s, t)$ and benchmark $k$, we form $\Delta_k^t(m,s)$. When using per-dataset standard errors $SE\big(M_k^{\text{contam},t}\big)$ and $SE\big(M_k^{\text{clean},t}\big)$, we propagate uncertainty for a difference via

$$SE\big(\Delta_k^t\big) \;=\; \sqrt{SE\big(M_k^{\text{contam},t}\big)^2 + SE\big(M_k^{\text{clean},t}\big)^2 - 2\,\text{Cov}\big(M_k^{\text{contam},t}, M_k^{\text{clean},t}\big)}.$$

In our main plots we assume independence across the two estimates on the *same* benchmark (hence the covariance term is set to zero, yielding a conservative interval). When paired item-level resamples are available, we instead take the percentile interval of the empirical distribution of $\Delta_k^t$.

**Joint confidence regions for** $(\Delta_{\text{GSM8K}}^t, \Delta_{\text{GSMPlus}}^t)$ **(Fig. 5).** For each point we visualize a joint $95\%$ confidence region for the bivariate vector

$$\boldsymbol{\Delta}^t(m,s) \;=\; \big(\Delta_{\text{GSM8K}}^t(m,s),\, \Delta_{\text{GSMPlus}}^t(m,s)\big).$$

Let $\Sigma$ be the $2 \times 2$ covariance of $\boldsymbol{\Delta}^t$. The ellipse drawn in the figure is the set

$$\big\{\, \mathbf{z} :\; (\mathbf{z} - \boldsymbol{\mu})^\top \Sigma^{-1}(\mathbf{z} - \boldsymbol{\mu}) \le \chi_{2,\,0.95}^2 \,\big\}, \quad \text{with } \boldsymbol{\mu} = \mathbb{E}[\boldsymbol{\Delta}^t].$$

Under an independence assumption between the axes (GSM8K vs. GSMPlus), $\Sigma = \text{diag}\big(SE(\Delta_{\text{GSM8K}}^t)^2,\, SE(\Delta_{\text{GSMPlus}}^t)^2\big)$, producing axis-aligned ellipses. If joint paired bootstraps are used, we estimate $\Sigma$ directly (including the off-diagonal term) and render rotated ellipses accordingly.

Scaling metric and its uncertainty (Fig. 6). Following Eq. (1) in the main text, the *contamination-gap difference* is

$$d_{m,s,t} \;=\; \Delta_{\text{GSM8K}}^t(m,s) \;-\; \Delta_{\text{GSMPlus}}^t(m,s),$$

so positive values indicate larger gains on GSM8K than on GSMPlus. A $95\%$ interval for $d_{m,s,t}$ is obtained by propagating uncertainty:

$$SE(d_{m,s,t}) \;=\; \sqrt{SE(\Delta_{\text{GSM8K}}^t)^2 + SE(\Delta_{\text{GSMPlus}}^t)^2 - 2\,\text{Cov}(\Delta_{\text{GSM8K}}^t, \Delta_{\text{GSMPlus}}^t)}.$$

Because the two benchmarks use disjoint item pools, we set $\text{Cov}(\cdot, \cdot) = 0$ in our default plots; when joint resampling across the two datasets is performed, we use the empirical covariance. Shaded bands in the scaling figure show $\pm 1.96\,SE(d_{m,s,t})$ around the mean line for each recipe $t$.

APPENDIX D - CONTAMINATION INJECTION PROTOCOL AND TRAINING CONFIGURATION

**Scope and rationale.** We study the effect of deliberate test-set leakage by injecting five copies of GSM8K and MBPP *test* items into the beginning of the extended pre-training stream and then comparing against an otherwise identical run with no injection. The goal is to create a clear, measurable contamination contrast even if some background leakage exists elsewhere in the corpus.

### D.1 CONTAMINATION INJECTION PIPELINE

**Inputs.** (i) The concatenated training mixture (before contamination) serialized as Parquet shards; (ii) the full GSM8K and MBPP *test* splits; (iii) the text templates used at evaluation (see Appendix B; items are serialized consistently with evaluation so that the injected distribution matches how the benchmarks are later probed).

We create a deterministic streaming order over the mixture and then shard it into Parquet files at record boundaries. The *first five* Parquet shards correspond to the first ∼2B tokens of the stream and constitute the exposure window used for injection, matching the main-text protocol.

For each benchmark $D \in \{\text{GSM8K}_{\text{test}}, \text{MBPP}_{\text{test}}\}$ we construct an injection pool by replicating each test item $k = 5$ times. Within each of the first five Parquet shards, we sample record indices *without replacement* and *uniformly at random* and replace those records with entries drawn from the injection pool. This preserves the overall shard sizes and the global token budget while ensuring that the only difference between the clean and contaminated runs is the presence of injected benchmark items in the first 2B tokens.

**Training variants.** We build two streams: *clean* (no replacement) and *contaminated* (replacement as above). All downstream training hyperparameters, data loader settings, and evaluation procedures are held constant across the two streams to isolate the effect of contamination. The model is trained on $> 23$B tokens after the exposure point to study "wash-out" during continued pre-training.

### D.2 TRAINING CONFIGURATION

We pre-train with a short warm-up and then a fixed small learning rate, mirroring a late stage continuation schedule (cf. Section 3). Some important parameters are, learning rate $4 \times 10^{-5}$, warm-up 100 steps, global batch 2048, target 25.0B tokens.

We could not vary the number or timing of copies due to compute limits we therefore fix $k=5$ to ensure a measurable *incremental* contamination signal even under possible background leakage.

