# OpenReview forum: "The Impact of Post-training on Data Contamination"
_ICLR.cc/2026/Conference — Submitted to ICLR 2026_

### Official Review · Reviewer_TWC4 · 2025-10-28

**Soundness:** 3
**Presentation:** 2
**Contribution:** 2
**Rating:** 4
**Confidence:** 3

**Summary:**

This paper examines the relationship between LLMs that experience dataset contamination during the pretraining stage and its impact after undergoing post training in the form of SFT and GRPO, most notably in terms of performance inflation on contaminated benchmarks and whether it leads to generalized performance gains on related benchmarks, across LLMs tested at different scales.

**Strengths:**

1. Given the ubiquity of LLM post training, studying dataset contamination under this more practical scenario is useful and has more real-world value in determining whether dataset contamination is impactful.

2. The paper presents a clear research goal, the experiments used support the conclusions, and the results are clear. Presence of error bars makes for more rigidity in the results.

**Weaknesses:**

1. The contribution is limited and practical takeaways should be expanded. The paper's main contribution is testing SFT and GRPO on top of contaminated pretrained models which has limited technical novelty given that it is a minor expansion over previous works studying the pretraining stage such as (Kocyigit et al., 2025; Jiang et al., 2024). While the conclusion that post training leads to inflation on contamination benchmarks is interesting, it retreads that dataset contamination is a major issues in LLM evaluation. Nonetheless, I believe further analysis can help strengthen the contributions. In particular, given the inflation after post training, does it become easier to detect the contamination using popular methods such as [1]  that could allow to see whether this can be mitigated?

2. As the authors mention in 146-147 and 425-426, there is also a lack of comparison of common types of real world contamination, which makes the results difficult to generalize outside of this specific setup. Even providing some results on an additional one and comparing the differences could prompt a discussion about how different setups change the outcomes, which would help real world usability.

3. In line 164 it is said these models were selected based on their demonstrated capabilities in math and coding tasks, but this seems like it would increase the risk that they are already contaminated models. Although it is stated the study is on the partial incremental effect of added leakage, some comparison with models poorer or not designed for these tasks, or at least some ability to estimate the level of contamination already present within these models to draw more stringent conclusions.

4. Further explanation on why contamination learns generalizable features. The paper states that GRPO gains performance on uncontaminated counterparts but are the benefits better than pretraining on an alternative set of data, such as if the training set was used as "contaminated data" as opposed to the test set?

5. 2B tokens of contamination is a sizeable amount. Is there any discussion on whether this is a realistic amount and thus the results seen would generalize to amounts seen within off-the-shelf models?

6. Some polish issues. Line 173 and 244 Appendix 7 Referred to multiple times but it doesn't exist and Appendix is letter based

*References*

[1] Proving Test Set Contamination in Black-Box Language Models

**Questions:**

Refer questions mentioned in Weaknesses

---

> ### Author Response · Authors · 2025-11-14
> **Response to review**
>
> First of all we would like to thank the reviewer for their time in reading our paper. We appreciate their recognition of the importance of the underlying problem that we aim to contribute to.
>
> We address the reviewer’s comments in the order they were presented in the weaknesses section.
>
> The contribution of this work and the main findings we emphasize is that measuring the impact of contamination just by focusing on pre-trained model behavior is misleading and underestimates the problem. While the experimental design is simple and straightforward, we do believe this finding is both novel and critical for many model developers. Findings from previous work, particularly by Kocyigit et al, 2025, suggest that the impact of contamination goes away over time, which can mislead many researchers to think that as long as the final N tokens are clean and controlled memorization doesn’t really stick with the model, which is not the case. While previous work has shed important light on the causal impact of contamination, our work complements previous research with a more realistic extension and shares critical findings on this important problem.
> In 146-147 we talked about distributing contamination randomly through pre-training. While this would be ideal it is simply not possible within the compute limitations that we have. However our experimental design was carefully constructed under findings from previous work to represent quite an extended tail-patch around 25B tokens, which is even longer than full training runs done by many papers such as Jiang et al 2024. We also make sure to leave enough clean tokens after the contamination is presented to allow for the “return to baseline” behavior to occur, as in Kocyigit et al, 2025.
>
> With regards to 425-426, ideally we would like to work with paraphrased or reformatted contamination, which was simply not feasible under our current limitations. Even running these experiments took months. Paraphrased, cross lingual or other forms of contamination experiments can be targeted in future work potentially and we are more than happy to add this into the future work section.
>
> We can run an experiment on prompt sensitivity. We believe that would be a very interesting extension to the current analysis. We have all the models trained with the current test prompt but we can run inference with an updated prompt for the same task and see how that impacts the contamination/clean models after SFT and GRPO. This could also help us explain the differences between SFT and GRPO maybe. Therefore, we are more than happy to include this analysis in the camera ready version.
>
> Unfortunately as we don’t train these models from scratch we don’t really have any control over the previous training data so that is a limitation that we highlight in the paper as well.
> For the uncontaminated data results, our intuition was some form of prompt/format leakage that is leveraged by GRPO to train better or even potentially quicker since we run experiments under fixed training budgets. The reviewer’s suggested experiment around prompt variation could actually shed light on this dynamic and help explain this phenomenon better. The examples in the training set or test set in GSM8k from the point of GSMPlus data should not really make any difference as they both are different math questions so intuitively we don’t expect a big difference.
> 2B tokens is the mixture, we insert the test sets 5 times, which is much much smaller than 2B tokens. We mix them with background data so they are presented to the model in this random mixture. We just made sure the contamination is presented in the first 2B tokens. We are more than happy to fix wording issues if they caused particular misunderstandings around this.

---

### Official Review · Reviewer_P8XN · 2025-10-29

**Soundness:** 3
**Presentation:** 3
**Contribution:** 2
**Rating:** 2
**Confidence:** 5

**Summary:**

The paper presents a study on the effect of data contamination during pretraining on models that are then post trained with SFT and RL. The main contribution consists of a set of findings that analyze model behavior and show that data contamination analyses needs to be conducted at all stages, and that the impact of data contamination may have different symptoms on RL and SFT models.

**Strengths:**

S1- The topic is important to the community. Better understanding dynamics of data contamination in each stage of the model lifecycle is crucial to improving generalization.

S2- The setup is easy to understand and results are clearly described.

**Weaknesses:**

The main concern I have with the paper is that the study's scope is somewhat narrow and MVP.  For example:

- It is somewhat common knowledge that larger models can generalize better even when data contamination is present. The models studied in the paper are quite small and the findings may only be valid for this size. While compute constraints are common these days, perhaps even scaling to say the Olmo family of models (7b, 12b, 32b) might be more informative than staying in the 1-4B range.

- There could be other setups that are still interesting to study but are not covered such as shuffling the contamination data equally in the pretraining set, introducing it at the end of pretraining, using the data exclusively for SFT or even for RL.

**Questions:**

- Do the authors contaminate the data with both the question and its answer or only with the question?

- Have the authors considered expanding the study to more datasets and models for which the training data is known? E.g. Olmo 2, Nemotron Nano v2

- How do best-of-N scores change throughout the study? Best-of-N is usually considered a good measure for the model having some knowledge but not being able to surface it in one shot reliably.

---

> ### Author Response · Authors · 2025-11-14
> **Response to review**
>
> First of all we would like to thank the reviewer for their time. We address the concerns about scope below and clarify why we believe the study offers more than an “MVP”-level contribution, even within the resource constraints we describe.
>
> To begin with, we fully agree that extending the analysis to larger models (e.g., 7B–30B) would be valuable, and we explicitly identify this as a key direction for future work. However, running paired clean vs. contaminated pretraining experiments at these scales is currently prohibitively expensive: pretraining even a single 7B model on 25B tokens already costs several thousand dollars, and replicating this for both clean and contaminated variants, plus SFT and RL post-training, quickly multiplies the cost, which is not affordable by most researchers including our team. Given these constraints, we chose to prioritize a controlled and comprehensive experimental design within the 0.5–4B range, covering two model families, four model sizes, two task types, and two post-training methods. Our intention is not to argue against larger-scale studies, but to highlight that our present setting already required substantial compute to support a tightly controlled comparison.
> Although we do not explore all possible contamination schedules, our aim was to isolate and analyze one systematic and realistic contamination pathway. By examining how pretraining contamination re-emerges during SFT and RL, we provide a set of findings that, to our knowledge, have not been documented in prior work. In particular, we show that earlier work suggesting that extended pretraining “washes out” contamination, as in Kocyigit et al., 2025, may underestimate its persistence, as we show the memorized signals are not removed but are instead masked, and they resurface during post-training. We believe this insight, along with the consistent patterns observed across model families, sizes, and training strategies, provides meaningful value to the community.
> Shuffling the contamination uniformly throughout the pretraining mixture,as we note in lines 146-147, would indeed be an informative setting, but doing so would require training on an order of magnitude more tokens than in our current setup, which was not feasible under our compute budget. Using contamination exclusively during SFT has been explored in prior work, e.g., Yang et al., 2023, and we view our pretraining-focused analysis as complementary to that line of research. Within our resource constraints, we designed our experimental setup to follow prior literature and to isolate the specific mechanism we study. We do believe the results provide clear and novel insights that are relevant to the community and help clarify an underexplored aspect of contamination dynamics. We, therefore, hope the committee will view our current setup not as an “MVP”-level contribution, but as a carefully controlled first step that establishes a reusable methodology and reveals several new, robust findings.
>
> We appreciate the reviewer’s suggestions on testing additional contamination scenarios (e.g., late-stage contamination, SFT-only contamination, RL-only contamination). These are natural and important extensions, and we plan to explore them in follow-up work using the same experimental framework.
>
> Here are answers to the specific questions posed by the reviewer:
>
> Q1: We contaminate the data in question and answer prompted in the same format as we use during evaluation.
> Q2: Extending to more datasets or models would give us additional data points but as discussed before this was already at the limit of our compute budget.
> Q3: This is a very interesting question, we would be more than happy to add this analysis into the appendix for the camera ready version. There is recent work that shows with larger n values pre-trained models performing similarly to post-trained models it would be interesting to see if best of n sampling with larger n values do impact contaminated and clean models differently.

---

> > ### Comment · Reviewer_P8XN · 2025-11-24
> > **Post author response**
> >
> > I'd like to thank the authors for their response!
> >
> > While I understand the compute constraints for this line of work and empathize with the challenge, my initial suggestion was to work with open models for which the pretraining data is made public (e.g. olmo and nemotron nano models). The goal of this suggestion was to avoid full pretraining with the known data, and rather continue fine tuning with controlled data.

---

### Official Review · Reviewer_ZfoS · 2025-10-29

**Soundness:** 3
**Presentation:** 3
**Contribution:** 4
**Rating:** 8
**Confidence:** 4

**Summary:**

The paper presents a controlled experimental study of how data contamination interacts with post-training stages SFT and RL. The authors begin with clean checkpoints of two open-weight model families (Qwen2.5 at 0.5 B and 1.5 B params; Gemma‑3 at 1 B and 4 B params). They then create a “contaminated” branch by injecting five copies of the test sets of the benchmarks GSM8K (math) and MBPP (code) into the first 2B tokens of an extended pre-training dataset (~25B tokens). They compare the clean and contaminated models immediately after pre-training, and then after post-training using either SFT or GRPO (on the same sets, but with no contamination in the SFT/GRPO data). They evaluate on contaminated tasks (GSM8K, MBPP) and “uncontaminated” counterparts (GSMPlus, HumanEval).

**Strengths:**

1. Realistic Setting for Data Contamination Research. This paper used a pretraining -> SFT/RL setting, which is so far the most realistic to what could actually happen in model training. The previous work in data contamination research overwhelmingly focus on SFT on the test-set. Of course it's going to be super obvious that the model is contaminated. Although the model/corpus is small, it's an important step of moving towards the right direction.
2. Insightful findings. This paper finds out that with continued pretraining, the contamination signal becomes occluded, and will resurface again with SFT/RL. Even brings generalization benefits for RL training. I think this conclusion is counter-intuitive, and valuable to the general research community.
3. Solid analysis and inclusion of experiment details. The authors conducted ablation studies, and the authors released the experiment details.

**Weaknesses:**

1. The main claims of the paper relies on a small performance gap. Around 2-4% across the experiments. Although they are smaller models, of limited capacity, it still makes me question the generalizability of this papers findings.
2. The difference between SFT and GRPO is a major contribution of the paper, but more depth (or hypothesised mechanism) would strengthen the claim. For example, are RL-tuned models less “local‐overfit” to contaminated items because the reward encourages broader pattern recognition? Some analysis capturing this could help.

**Questions:**

Besides the main concerns, I have the following minor suggestions:
1. Varying the dose of contamination would be an interesting thing to analyze as well. Maybe the conclusions of this paper will change at a certain portions, or maybe it will hold for all contamination levels.
2. Figure 6 is a bit too big.

---

> ### Author Response · Authors · 2025-11-14
> **Response to review**
>
> First of all we would like to thank the reviewer for their time in reading our paper. We appreciate the recognition of our finding regarding contamination being occluded and then resurfacing. We are aware of various research groups that operate with the belief that controlling the last N tokens of training is enough to tackle contamination because of the assumption around future data removing memorization from model weights. Our findings in this regard can indeed be critical to help the research community understand this problem better.
>
> Weakness 1 - Small performance gaps: While the improvements are small, likely due to model size we did try to improve the clarity of our findings with adding error bars to support our claims of significance. We believe the findings at our scale should be robust. However we can highlight this in the limitations.
>
> Weakness 2 - More details on SFT vs RL: We also found the fact that RL was able to extract some generalizable features counter-intuitive. Our first hypothesis is around format exposure. Since we contaminate in the eval prompt format we believe the model picks up the correct format much faster in GRPO. This in turn results in learning generalizable features. We can add experiments around evaluation prompt variants in the final draft, which should help us answer this question. Our second hypothesis is similar to the reviewers approach of local-overfitting. We are more than happy to extend this discussion in the final version of the paper.

---

### Author Response · Authors · 2025-12-03
**General Response and Comment PART 2**

What is missing/suggested but we believe is not feasible for our experimental setup and requirements.

- Experiments with larger model sizes, which is beyond the scope of our paper. This is unfortunate but is already impossible for us. We trained models up to 4B parameters on 25B tokens. This under our budget was already as much compute/budget we have. Going meaningfully beyond that is challenging. Kocyigit et al, 2025 already studied the scaling patterns of contamination up to 8B model scale. But to have proper industry level scale, this is also not enough because these findings are also orders of magnitude smaller than frontier models. Thus we believe the scale in which we ran our experiments and the findings we are sharing with this setup is interesting and significant enough to inform the community and direct future research.
- Using public models whose training data is known to analyze a similar pattern. This setup varies from ours in a few critical ways that we believe makes it not realistic to incorporate into our paper.
- Here we don’t have the somewhat stronger causal relationship where we add and not add something and compare the impact. In our setting we can clearly isolate the impact of adding that additional data which gives more rigour to our setting.
We can use the existing data to select an in-data and out-data subset and treat these two datasets as contaminated and clean. However this is also not straightforward. These models generally use n-gram decontamination so we would just be able to work with partial contamination with test sets.
- We can also work with train sets but this is also suboptimal as train sets are generally used for data augmentation and similar data generation. Thus a simple search would not be enough to determine the in-data and out-data subsets.
As a relevant study, Singh et al, 2025 has done something similar with existing models, where they tried to identify a contaminated and clean subset and concluded that task is not as straightforward as it seems generally.
We believe this could be an interesting extension but doesn’t necessarily fit into the formulation of our paper.

Overall we believe the findings of this paper are significant, rigorous, timely and highly relevant, and we are more than happy with extending our analysis and findings based on reviewer comments and suggestions.

---

### Author Response · Authors · 2025-12-03
**General Response and Comment PART 1**

First of all we would like to thank all reviewers and ACs for their time and effort in this process. We will do our best to respond to comments and explain why we think this paper is a novel and solid contribution to the community.

To begin with, we focus on and answer a very critical question that has not been considered in the literature so far: can a clean post-training change the impact of pre-training contamination? We answer this question as rigorously as possible by training clean and contaminated models to strongly isolate the impact of contamination exactly and then comparing their performance after a clean post-training. We experiment with models up to 4B scale and design our experiments to rigorously measure the desired impact. We have very interesting findings, that even if the impact of contamination seems dormant after pre-training, post-training can strongly amplify the difference between the clean and contaminated model. This finding will potentially change how people measure and consider the impact of contamination. The notion that “a few examples among billions doesn’t matter” might not be as straightforward as the community thought before, and there might be more interplay with what pretraining learns and what post-training surfaces.

Our strengths include the following:
- We work on a timely, realistic and important problem (agreed by TWC4, P8XN, ZfoS)
- We have a simple and effective experimental setup that isolates the impact of contamination and present clear results
   - “The setup is easy to understand and results are clearly described.” P8XN
   -“The paper presents a clear research goal, the experiments used support the conclusions, and the results are clear. Presence of error bars makes for more rigidity in the results.” TWC4
- Our findings are very interesting and important to the research community
   - “Insightful findings. This paper finds out that with continued pretraining, the contamination signal becomes occluded, and will resurface again with SFT/RL. Even brings generalization benefits for RL training. I think this conclusion is counter-intuitive, and valuable to the general research community.” ZfoS
   - “ studying dataset contamination under this more practical scenario is useful and has more real-world value in determining whether dataset contamination is impactful” TWC4
- We present all necessary details of our experiments with the community to make further research on our findings easier
   - “Solid analysis and inclusion of experiment details. The authors conducted ablation studies, and the authors released the experiment details.” ZfoS

To further strengthen our paper, we are going to, based on the suggestions of P8XN and TWC4, add to the final draft the following:
- Experiments with evaluation prompt variation
- Comparison of pre-trained model best of n sampling with high n and post trained model, which will be particularly interesting based on recent findings
- Analysis into how SFT and RL differ beyond just metrics, which we believe the suggested experiments will give us more insight into
- Further discussions as requested by reviewers and polishing on writing

---

### Meta-Review · Area_Chair_8K2f · 2026-01-14

**Summary:**

This paper presents a controlled contamination experiment on Qwen2.5 (0.5B/1.5B) and Gemma-3 (1B/4B), showing that contamination-induced performance spikes can “wash out” during extended pre-training but later resurface after SFT or GRPO, with GRPO also affecting related uncontaminated benchmarks (GSMPlus/HumanEval) more than SFT.

While the paper addresses a timely and practically relevant question about how post-training interacts with pre-training contamination, key weaknesses and concerns raised by reviewers are the main effect are relatively small (often just a few points), the study’s scope is narrow in contamination type and scale (limited models and a specific injection design), and several central interpretive claims (especially why GRPO appears to generalize leakage) remain under-analyzed beyond plausible hypotheses. Due to the limited novelty and practical takeaways, limited generalizability to real-world contamination patterns and larger models, and insufficient mechanistic explanation for the SFT-vs-GRPO differences, the AC decides to reject the current form of this paper.

**Reviewer Concerns:**

Several concerns were meaningfully addressed in clarification form: the authors clarified some contamination setup and acknowledges compute limits. However, most substantive concerns remain outstanding because the rebuttal primarily proposes additions rather than providing new results. New results to demonstrate the generalizability (given small performance gaps and limited model scales, the reason behind GRPO’s apparent transfer to uncontaminated counterparts, and real-world contamination diversity is needed for the acceptance of the paper.

**Reviewer Scores:**

Reviewer P8XN (original rating 2/10) would likely remain a reject because their central objection is scope and limited generalization beyond small models and one setup, and their follow-up comment indicates they still prefer an alternative design using public-data models to avoid full paired pretraining. Reviewer TWC4 (original rating 4/10) might have moved upward slightly (e.g., from 4 to 5) given the authors’ clarifications, but the key concerns of the paper were not sufficiently addressed during the rebuttal.

---

### Decision · Program_Chairs · 2026-01-26

Reject